# MARLlib: Extending RLlib for Multi-agent Reinforcement Learning

## Abstract

Despite the fast development of multi-agent reinforcement learning (MARL) methods, there is a lack of commonly-acknowledged baseline implementation and evaluation platforms. As a result, an urgent need for MARL researchers is to develop an integrated library suite, similar to the role of RLlib in single-agent RL, that delivers reliable MARL implementation and replicable evaluation in various bechmarks. To fill such a research gap, in this paper, we propose Multi-Agent RLlib (MARLlib), a comprehensive MARL algorithm library that facilitates RLlib for solving multi-agent problems. With a novel design of agent-level distributed dataflow, MARLlib manages to unify tens of algorithms, including different types of independent learning, centralized critic, and value decomposition methods; this leads to a highly composable integration of MARL algorithms that are not possible to unify before. Furthermore, MARLlib goes beyond current work by integrating diverse environment interfaces and providing flexible parameter sharing strategies; this allows to create versatile solutions to cooperative, competitive, and mixed tasks with minimal code modifications for end users. A plethora of experiments are conducted to substantiate the correctness of our implementation, based on which we further derive new insights on the relationship between the performance and the design of algorithmic components. With MARLlib, we expect researchers to be able to tackle broader real-world multi-agent problems with trustworthy solutions. Our code[1] and documentation[2] are released for reference.

## 1 Introduction

Multi-Agent Reinforcement Learning (MARL) is a prosperous research field that has many real-world applications and holds revolutionary potential for advanced collective intelligence [6, 38, 36]. Existing work [2, 33, 5] has shown that agents are able to learn strategies that could outperform human experts and help guide human's decision-making process in reverse. Significant as these outcomes are, the algorithm implementations are always task-specific, making it hard to compare algorithm performances, observe algorithm robustness across tasks, or use them off the shelf. Thus, developing a commonly-acknowledged baseline implementation and a unified tool suite for MARL research is in urgent demand.

While single-agent RL has witnessed successful unification for both algorithms (e.g. SpinningUp [1], Tianshou [35], RLlib [19], Dopamine [7] and Stable-Baselines series [10, 12, 25]) and environments (e.g. Gym [4]), multi-agent RL has unique challenges in building a comprehensive and high-quality library. Firstly, there exist diverse MARL algorithm pipelines. MARL algorithms diverge in learning targets such as working as a group and learning to cooperate, or competing with other agents and finding a strategy that can maximize individual reward while minimizing others. Algorithms also have different restrictions on agent parameters sharing strategies, with HATRPO agents forced to not share parameters and MAPPO capitalizing on sharing. Different styles of central information utilization such as mixing value functions (e.g. VDN [30]) or centralizing value function (e.g. MADDPG [20]) introduce extra challenge on algorithm learning style unification. Existing libraries such as EPyMARL [23] attempt to unify MARL algorithms under one framework by introducing independent learning, centralized critic, and value decomposition categorization but still lack the effort to address

---

[1] https://github.com/ICLR2023Paper4242/MARLlib
[2] https://iclr2023marllib.readthedocs.io/

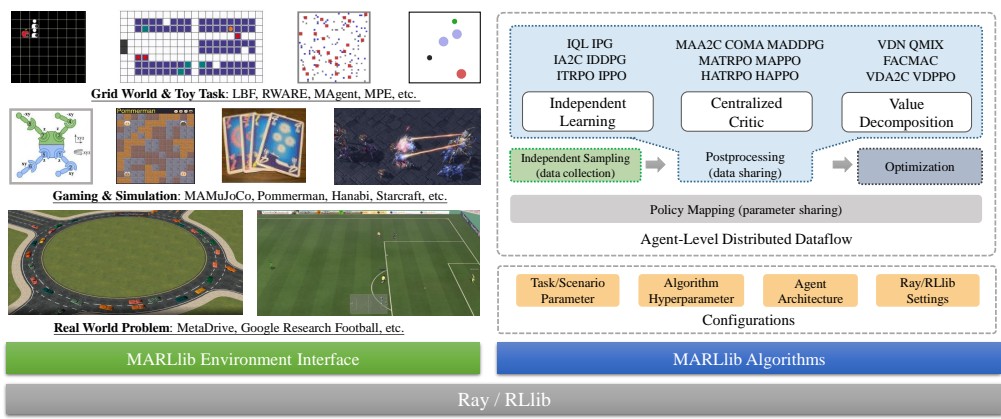

Figure 1: An overview of Multi-Agent RLlib (MARLlib). MARLlib unifies environment interfaces to decouple environments and algorithms. Beyond, it unifies independent learning, centralized critic, and value decomposition algorithms with an agent-level distributed dataflow, and allows flexible parameter sharing by means of policy mapping. The whole pipeline can be fully determined by configuration files. To our best knowledge, with the widest coverage of algorithms and environments, MARLlib is one of the most comprehensive MARL research platform.

all the problems above. The diversity of MARL algorithms is still a huge challenge for unification. Secondly, various multi-agent environment interfaces are mutually inconsistent, as they are originally designed to fit the task nature (e.g. asynchronous interaction is used in Hanabi, action masks are provided as additional information in SMAC [28], local observation and global state are mixed in MAgent [39]). The inconsistency hinders a directly unified agent-environment interaction processing and results in the issue of coupling between algorithm implementation and task environment; an algorithm implementation for one environment can not be directly applied to another due to interface changes. While PettingZoo [32] builds a collection of diverse multi-agent tasks, it is inconvenient for CTDE-based algorithm implementation as important information such as global state and action mask is not explicitly provided. Towards the inconsistency problem, other work, such as MAPPO benchmark [37], provides each environment with a unique runner script. Nevertheless, this solution creates hurdles for long-term maintenance as well as uneasiness for new task extensions.

To address the above challenges in one work, we build a new library called **MARLlib** based on Ray[22] and RLlib. By inheriting core advantages from RLlib and providing the following four novel features, MARLlib serves as a comprehensive platform for MARL research community.

**1. Unified algorithm pipeline with a newly proposed agent-level distributed dataflow**: To unify algorithms under diverse MARL topics and enable them to share the same learning pipeline while preserving their unique optimization logics, we construct MARLlib under the guidance of a key observation: all multi-agent learning paradigms can be equivalently transformed to the combination of single-agent learning processes; thus each agent maintains its own dataflow and optimizes the policy regardless of other agents. With this philosophy, algorithms are implemented in a unified pipeline to tackle various types of tasks, including cooperative (team-reward-only cooperation), collaborative (individual-reward-accessible cooperation), competitive (individual competition), and mixed (teamwork-based competition) tasks. We further categorize algorithms based on how they utilize central information, thereby enabling module sharing and extensibility. As shown in Figure 1, MARLlib manages to unify tens of algorithms with the proposed agent-level distributed dataflow, validating its effectiveness.

**2. Unified multi-agent environment interface**: In order to fully decouple algorithms from environments, we propose a new interface following Gym standard, with a data structure design that is compatible with most of the existing multi-agent environments, supports asynchronous agent-environment interaction, and provides necessary information to algorithms. To show the advantage of our interface design, MARLlib supports ten environments (SMAC [28], MAMuJoCo [24], GRF [16], MPE [20], LBF [23], RWARE [23], MAgent [39], Pommerman [27], MetaDrive [18], and Hanabi [3]) picked from the zoo of multi-agent tasks because of their inter-diversity, covering various task

Table 1: A comparison between current MARL libraries and our MARLlib. (x) stands for the number of available algorithms. * denotes that the benchmark has a unique framework of its own.

| Library | Task Mode | Supported Env | Algorithm | Parameter Sharing | Async Sampling | Framework |
|---------|-----------|---------------|-----------|-------------------|----------------|-----------|
| PyMARL [11] | cooperative | 1 | Independent Learning (1) Centralized Critic (1) Value Decomposition (3) | full-sharing | | * |
| PyMARL2 [13] | cooperative | 1 | Independent Learning (1) Centralized Critic (1) Value Decomposition (9) | full-sharing | | PyMARL |
| MARL-Algorithms [21] | cooperative | 1 | CTDE (6) Communication (1) Graph (1) Multi-task (1) | full-sharing | | * |
| EPyMARL [23] | cooperative | 4 | Independent Learning (3) Centralized Critic (4) Value Decomposition (2) | full-sharing non-sharing | | PyMARL |
| MAlib [40] | self-play | 2 + PettingZoo [32] OpenSpiel [17] | Population-based (9) | full-sharing group-sharing non-sharing | ✓ | * |
| MAPPO benchmark [37] | cooperative | 4 | Multi-agent PPO (1) | full-sharing non-sharing | ✓ | pytorch-a2c-ppo-acktr-gail [15] |
| **MARLlib** | cooperative collaborative competitive mixed | 10 + PettingZoo | Independent Learning (6) Centralized Critic (7) Value Decomposition (5) | full-sharing group-sharing non-sharing | ✓ | Ray [22]/RLlib [19] |

settings in MARL, including differences of task mode, observation dimension, action space property, agent-environment interaction style, etc.

**3. Effective policy mapping**: Flexible parameter sharing is the key to enabling one algorithm to tackle different task modes. To reduce the manual effort on regulating policies assignment, MARLlib provides three parameter sharing strategies, namely full-sharing, non-sharing, and group-sharing, by implementing the policy mapping API of RLlib. By simply changing the configuration file, users can switch among these strategies regardless of algorithms or scenarios. Therefore, although policy mapping controls the correspondence between policies and agents throughout the pipeline, it is fully decoupled from algorithms and environments, enabling further customization of sharing strategies.

**4. Exhaustive performance evaluation**: We run all suitable algorithms on 23 selected scenarios of five most common and diverse environment suites under four random seeds on average, which sums up to over one thousand experiments in total. The empirical results not only substantiate the correctness of MARLlib, but they also serve as a useful and trustworthy reference for the MARL research community as the comprehensiveness and fairness of comparison are guaranteed by the unified implementation approach. In addition, hyper-parameter tables are provided to ensure reproducibility. Based on these results, we derive key observations and analyze them in depth in Section 5.

With the above characteristics for a new MARL benchmark, MARLlib becomes one of the most general platforms for building, training, and evaluating MARL algorithms.

## 2  RELATED WORK

Building a unified platform for MARL research is meaningful yet challenging. MARL research has witnessed the development of algorithm library starting from a single task with a limited number of algorithms to more enriched tools and APIs, covering diverse tasks and advanced algorithms.

**PyMARL** [11] is the first and most well-known MARL library. All algorithms in PyMARL are built for SMAC [28], where agents learn to cooperate for a higher team reward. However, PyMARL has not been updated for a long time and can not catch up with the recent progress. To address this, the extension versions of PyMARL are presented including PyMARL2 [13] and EPyMARL [23].

**PyMARL2** [13] focuses on credit assignment mechanism and provide a finetuned QMIX [26] with state-of-art-performance on SMAC. The number of available algorithms increases to ten, with more code-level tricks incorporated.

**EPyMARL** [23] is another extension for PyMARL that aims to present a comprehensive view on how to unify cooperative MARL algorithms. It first proposed independent learning, value decomposition, and centralized critic categorization but is restricted to cooperative algorithms. Nine algorithms are implemented in EPyMARL. Three more cooperative environments LBF [8], RWARE [8], and MPE [20] are incorporated to evaluate the generalization of the algorithms.

All PyMARL-based libraries follow the centralized training decentralized execution setting as PyMARL. There also exist other MARL libraries that are built in different styles and serve their unique purposes.

**MARL-Algorithms** [21] is a library that covers broader topics compared to PyMARL including learning better credit assignment, communication-based learning, graph-based learning, and multi-task curriculum learning. Each topic has at least one algorithm, with nine implemented algorithms in total. The testing bed is limited to SMAC.

**MAPPO benchmark** [37] is the official code base of MAPPO [37]. It focuses on cooperative MARL and covers four environments. It aims at building a strong baseline and only contains MAPPO.

**MAlib** [40] is a recent library for population-based MARL which combines game-theory and MARL algorithm to solve multi-agent tasks in the scope of meta-game.

Existing libraries and benchmarks provide good platforms for developing and comparing MARL algorithms in different environments. However, there are essential limitations to the current work. Firstly, these work is limited in task coverage. As shown in Table 1, most existing work only supports one task mode. Moreover, the number of supported environments is insufficient for researching general MARL. Secondly, existing work pays little attention to how algorithms are organized but only focuses on implementing existing work. This results in poor extensibility and a bloated code structure. Our **MARLlib**, a comprehensive and unified library based on Ray and RLlib, effectively solves the dilemma by providing better algorithm unification and categorization, implementing more algorithms both in quantity and in diversity, covering four task modes, supporting ten environment suites, allowing flexible parameter sharing, and being friendly to different training demands.

## 3 MARLLIB ARCHITECTURE

In this section, we mainly explain how MARLlib addresses the two major challenges, namely the diversity of MARL algorithms and inconsistency of environment interfaces, with a newly proposed agent-level distributed dataflow, a unified agent-environment interface, and effective policy mapping.

### 3.1 AGENT-LEVEL DISTRIBUTED DATAFLOW FOR ALGORITHM UNIFICATION

A common framework to solve multi-agent problems is Centralized Training Decentralized Execution (CTDE), where agents maintain their own policies for independent execution and optimization, and centralized information can be utilized to coordinate agents' update directions during the training phase. Under this framework, existing libraries split the whole learning pipeline into two stages: data sampling and model optimization. In the model optimization stage, all data sampled in the data sampling stage are available to make the training centralized. However, in this way, choosing proper data and using these data to optimize the model are coupled in the same stage. As a result, extending an algorithm to fit other task modes (e.g. both cooperative and competitive) becomes more challenging and requires redesigning the whole learning pipeline.

MARLlib addresses this issue by equivalently decomposing the original grouped dataflow into agent-level distributed dataflow. Essentially, it takes every agent in multi-agent training as an independent unit during sample collection and optimization, but shares centralized information among agents during the `postprocessing` phase (`postprocessing` is a RLlib API for processing sampled data before model optimization; we enrich it to accommodate diverse algorithms) to ensure the equivalence. In `postprocessing`, agents share observed data (data sampled from the environment) and predicted data (actions taken by their policies or Q values) with others. All agents maintain individual data buffers, which store their experiences and necessary information shared by other agents. After entering the learning stage, no information sharing is needed among agents and they can optimize themselves independently. In this way, we *distribute* originally combined dataflow to agents and fully

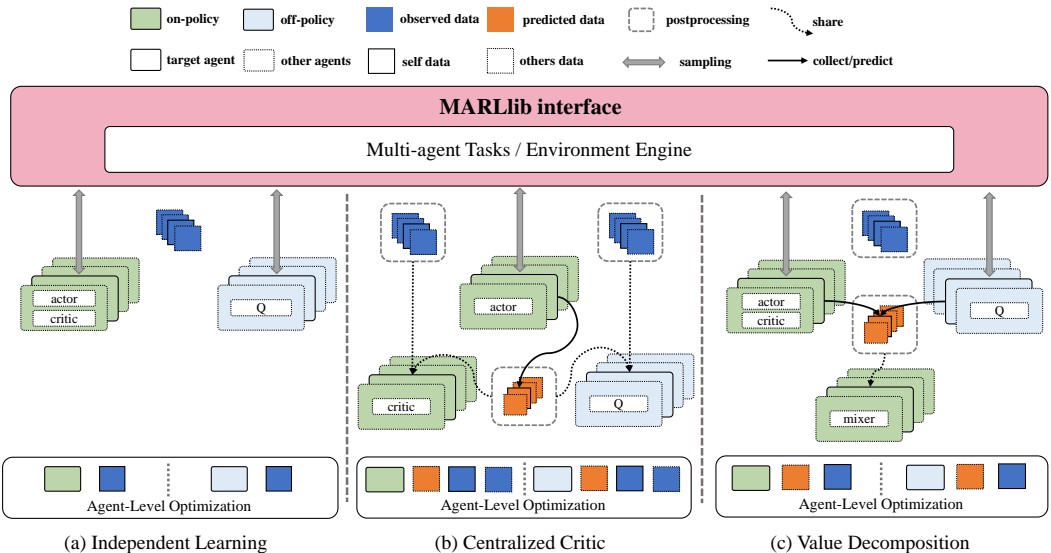

Figure 2: Agent-level distributed dataflow of MARLlib. Observed data refer to data sampled from environments, such as rewards or global states. Predicted data refer to data generated by agents, such as Q values or chosen actions. `postprocessing` refers to the data sharing process. Each agent maintains its own learning pipeline where collected data are used to optimize the policy of agent itself. Therefore, the dataflow is agent-level distributed. There are three types of dataflow, namely independent learning, centralized critic, and value decomposition [23], which are separated according to their central information utilization style. Independent learning algorithms (e.g. IQL [31]) are inherently distributed dataflow from the single-agent perspective and the information-sharing process is skipped, as shown in (a). For centralized critic algorithms (e.g. MAPPO [37]), central information, including both observed data and predicted data, is collected and shared in the `postprocessing` function before entering the training stage to ensure the distributed dataflow, as shown in (b). In value decomposition category (e.g. FACMAC [24]), predicted data from all agents must be shared, whereas the observed data is optional, depending on the algorithm's mixing function. The corresponding dataflow is shown in (c).

decouple data sharing and optimization, thereby allowing the same implementation to solve multiple modes of tasks.

Moreover, while all CTDE-based algorithms share similar agent-level dataflow in general, they still have unique data processing logic. Inspired by EPyMARL [23], we further classify algorithms into independent learning, centralized critic, and value decomposition categories to enable module sharing and extensibility. Independent learning algorithms let agents learn independently; centralized critic algorithms optimize the critic with shared information, which then guides the optimization of decentralized actors; value decomposition algorithms learn a joint value function as well as its decomposition into individual value functions, which agents then employ to select actions during execution. According to their algorithmic properties, we implement suitable data sharing strategies in `postprocessing` phase, as illustrated in Figure 2.

Therefore, preserving the unique properties of all algorithms, the agent-level distributed dataflow unifies diverse learning paradigms. Our implementation approach shows its value in unifying algorithms following CTDE in a single pipeline, which is capable of handling all task modes and achieving similar performance to the original implementation.

## 3.2 A UNIVERSAL INTERFACE FOR AGENT-ENVIRONMENT INTERACTION

In RL, Gym interface, `obs, reward, done, info`, is universally used. However, this standardized interface cannot be trivially extended to MARL. In MARL, multiple agents coexist, each having its own experience data. Extra information is sometimes available, such as action mask and global state. Depending on the task mode, the reward may be a scalar or a dictionary. Moreover, the

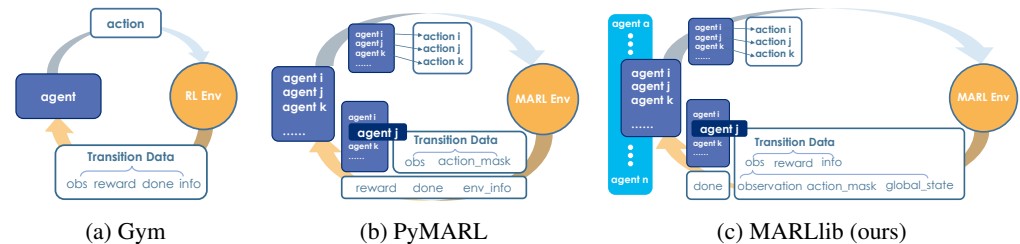

(a) Gym        (b) PyMARL        (c) MARLlib (ours)

Figure 3: Agent-environment interaction style comparison between Gym (standard RL), mostly used multi-agent library PyMARL, and MARLlib. MARLlib unifies diverse environment interfaces with a newly proposed Gym-style interface `obs, reward, done, info` and supports both synchronous and asynchronous sampling. `done` is a shared signal about termination, and the others are dictionaries containing agents-specific data indexed by agent id. In this figure, we organize the data by agent for better illustration.

agents may not interact with the environments synchronously, posing another challenge. Towards these issues, MARLlib unifies the multi-agent interface from two aspects.

Firstly, MARLlib unifies the interface data structure. MARLlib proposes a new interface following Gym standard, `obs, reward, done, info`, that is compatible with diverse multi-agent task settings in practice, covering both data structure and task modes. As illustrated in Figure 3c, in MAR-Llib, the observation returned from the environment is a dictionary with three keys: `observation`, `action_mask`, `global_state`. This design satisfies most circumstances and is compatible with RLlib's data processing logic. Other observation-related information is included in `info`. `reward` is a dictionary with agent id as the key. To accommodate cooperative tasks, scalar team reward is transformed into dictionary structure by copying it for agent number times. `done` is a dictionary containing a single key `"__all__"`, which is true only when all agents are terminated.

Secondly, MARLlib supports both synchronous and asynchronous agent-environment interaction. Existing MARL libraries before MARLlib, such as PyMARL (Figure 3b), do not support asynchronous sampling and mainly focus on synchronous cases. However, asynchronous agent-environment interaction is common in multi-agent tasks like Go and Hanabi. MARLlib supports synchronous and asynchronous agent-environment interaction, thanks to Ray/RLlib's flexible data collection mechanism: the data are collected and stored with agent id. Only when we receive the terminal signal `done` will all data be returned for subsequent usage. This sampling process is illustrated in Figure 3c.

### 3.3 EFFECTIVE POLICY MAPPING

In multi-agent scenarios, a proper parameter sharing strategy can improve the algorithm's performance. Important as it is, most existing work supports insufficient sharing modes and the implementation is repetitious — MAPPO benchmark rewrites everything for shared and separated settings, while EPyMARL repeats model structures to support both. In MARLlib, we support full-sharing (all agents share parameters), non-sharing (no agents share parameters), and group-sharing (agents within the same group share parameters) of parameters by implementing the policy mapping API of RLlib. Intuitively, it maps the virtual policies of agents to physical policies that are actually maintained, used, and optimized. Agents mapped to the same physical policy share parameters. As policy mapping is transparent to agents, they actually sample data and do optimization with the physical policies. Therefore, different types of parameter sharing can take place without affecting algorithm implementation. In practice, we only need to maintain a policy mapping dictionary for every environment with all the relevant information to support multiple sharing modes. A more customized parameter sharing strategy can be realized by revising the policy mapping API to suit the needs.

## 4 MARLLIB CONFIGURATION AND USAGE

MARLlib allows users to regulate the whole training pipeline by customizing configuration files, which is clean for usage and convenient for the experimental report. For a complete MARL pipeline, configurations of four different aspects are supported, including task configuration, algorithm con-

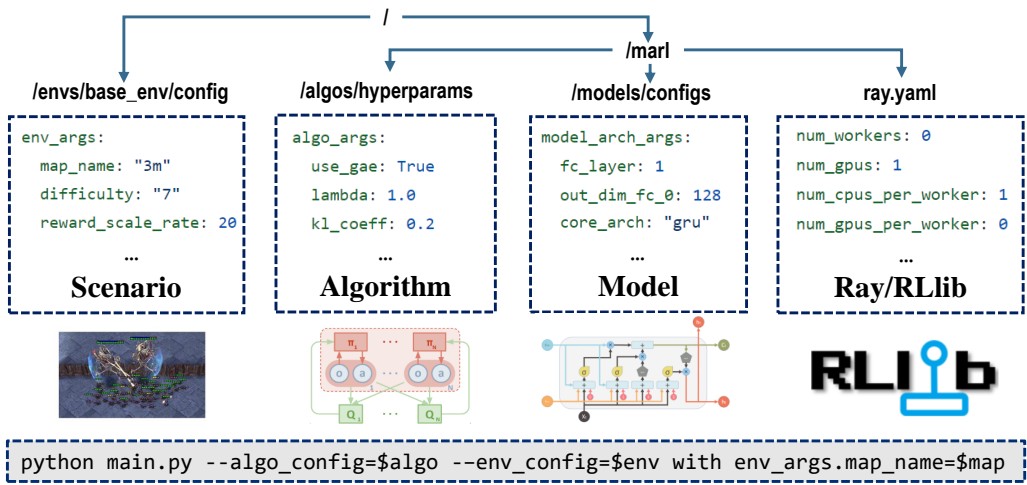

Figure 4: Functionalities and locations of configuration files in MARLlib. With these configuration files, MARLlib allows customization of environments, algorithms, models, and basic RLlib settings. When launching a training, only high-level choices of algorithms and tasks need to be specified, thereby avoiding long training commands and enabling easy experimental report.

figuration, agent model configuration, and basic training settings for Ray and RLlib. The contents and locations of them are shown in Figure 4. Task configuration maintains the key parameters of the task scenario when generating from the environment engine. For instance, the level of the enemy in SMAC can be decided by users. Algorithm configuration is in charge of the hyper-parameters of the algorithm's learning procedure. In addition, tricks can be turned on or off by simply changing the corresponding key values. As MARLlib aims at building a general benchmark, tricks that are only applicable for one task but not for others are not incorporated, which guarantees that MARLlib's algorithm configuration is valid for diverse multi-agent tasks. Agent model configuration is responsible for constructing the agent learning unit: a neural network. Basic training settings for Ray and RLlib control the computation resource allocation. By configuring the four aspects, the whole pipeline is determined, and the training can be launched by simply running the entrance script `main.py`. After the training begins, all configurations will be recorded. Referring to the record file can help reproduce the same performance and conduct a fair experiment.

## 5 BENCHMARKING RESULTS AND ANALYSIS

In this section, we evaluate 17 algorithms on 23 tasks of five MARL testing beds including SMAC [28], MPE [20], GRF [16], MAMuJoCo [24], and MAgent [39], which are chosen for their popularity in MARL research and their diversity in task modes, observation shape, additional information, action space, sparse or dense reward, and homogeneous or heterogeneous agent types. We report the mean reward of experiments under four random seeds, which sums up to over one thousand experiments in total. Experiments results are shown in Table 2 and Figure 5. Based on these results, we substantiate the quality of implementation and provide insightful analysis.

### 5.1 QUALITY OF IMPLEMENTATION

To show the correctness of MARLlib, we compare the performances of MARLlib implementation on SMAC to those reported by EPyMARL with the important hyper-parameters kept the same. Results of EPyMARL consume 40 million steps on on-policy algorithms and four million on off-policy algorithms. MARLlib only consumes half of them respectively, as we find it enough for training to converge. Even with fewer training steps, we match most of the performances reported by EPyMARL, as shown in Table 2. For all performance pairs available to compare, MARLlib attains similar results on 63% of them (the reward difference is less than 1.0), achieves superior results on 25% of them, and appears inferior on the rest 12%. Since every algorithm exhibits expected performances and for generality and stability, we do not rely on task-specific tricks. These experimental results substantiate the correctness of implementation. In this table, we also report for the first time the performances

Table 2: Algorithm performances (in reward) for cooperative tasks. Both discrete control tasks (MPE, SMAC, GRF) and continuous control tasks (MAMuJoCo) are covered. Among four environment suites, SMAC has two rows for each scenario. The first row is the performances reported by EPyMARL, and the second row is performances with MARLlib. For other environments, only MARLlib performances are included. **-** represents no data reported. Dark cells indicate the top two performances in each scenario.

| Env | Scenario | Independent Learning | | | | | Centralized Critic | | | | Value Decomposition | | | |
|---|---|---|---|---|---|---|---|---|---|---|---|---|---|---|
| | | IQL | IPG | IA2C | ITRPO | IPPO | MAA2C | COMA | MATRPO | MAPPO | VDN | QMIX | VDA2C | VDPPO |
| SMAC | 2s_vs_1sc | 16.72 | - | 20.24 | - | 20.24 | 20.20 | 11.04 | - | 20.25 | 18.04 | 19.01 | - | - |
| | | 16.09 | 20.07 | 20.07 | 20.16 | 20.18 | 20.09 | 10.32 | 20.23 | 20.21 | 16.3 | 17.25 | 15.61 | 20.24 |
| | 3s5z | 16.44 | - | 18.56 | - | 13.36 | 19.95 | 18.90 | - | 19.91 | 19.57 | 19.66 | - | - |
| | | 16.73 | 10.78 | 13.49 | 10.04 | 14.3 | 15.21 | 9.78 | 12.1 | 19.52 | 19.38 | 19.32 | 8.58 | 13.15 |
| | MMM2 | 13.69 | - | 10.70 | - | 11.37 | 10.37 | 6.95 | - | 17.78 | 18.49 | 18.40 | - | - |
| | | 12.08 | 9.21 | 10.17 | 8.04 | 10.37 | 16.08 | 6.7 | 7.62 | 16.86 | 19.31 | 18.34 | 2.72 | 9.31 |
| | 3s_vs_5z | 21.15 | - | 4.42 | - | 19.36 | 6.68 | 3.23 | - | 18.17 | 19.03 | 16.04 | - | - |
| | | 16.78 | 5.6 | 10.79 | 3.39 | 7.95 | 12.14 | 4.79 | 13.32 | 17.24 | 18.55 | 19.84 | 9.6 | 14.61 |
| MPE | simple_spread | -197.61 | -63.83 | -63.16 | -78.16 | -65.74 | -63.37 | -71.64 | -77.63 | -66.26 | -190.5 | -189.27 | -190.66 | -213.99 |
| | simple_speaker_listener | -44.07 | -261.65 | -29.06 | -50.17 | -38.29 | -27.76 | -67.6 | -44.01 | -34.41 | -35.26 | -25.68 | -54.37 | -64.61 |
| | simple_reference | -75.36 | -36.3 | -35.95 | -57.79 | -50.92 | -35.05 | -56.5 | -47.71 | -37.89 | -70.56 | -31.53 | -69.35 | -73.82 |
| GRF | pass_and_shoot | -0.17 | 0.6 | -0.03 | 0.6 | 0.5 | -0.02 | -0.01 | 0.48 | 0.74 | -0.06 | -0.24 | 0.05 | 0.01 |
| | run_pass_and_shoot | -0.15 | 0.07 | -0.07 | -0.05 | -0.07 | -0.05 | -0.03 | -0.02 | -0.03 | -0.24 | -0.11 | -0.09 | -0.13 |
| | 3_vs_1_with_keeper | 0.02 | 0.33 | 0.01 | 0.37 | 0.05 | 0 | 0.03 | 0.13 | 0.45 | -0.08 | -0.06 | 0 | 0 |

| | | IPG | IA2C | IDDPG | ITRPO | IPPO | MAA2C | MADDPG | MAPPO | HAPPO | FACMAC | VDA2C | VDPPO |
|---|---|---|---|---|---|---|---|---|---|---|---|---|---|
| MAMuJoCo | 2AgentAnt | 143.22 | -268.02 | 44.60 | 527.10 | -153.46 | 730.8 | 18.53 | -57.02 | 330.12 | -1224.6 | 449.19 | -98.74 |
| | 2AgentHalfCheetah | -133.06 | -457.11 | -197.85 | 1652.49 | -644.89 | -493.3 | -313.95 | -357.78 | 153.2 | -433.61 | -423 | -644.53 |
| | 2AgentWalker | 50.67 | 114.1 | 95.76 | 272.41 | 8.71 | 103.65 | 153.93 | -4.12 | 164.45 | -7.88 | 125.49 | -3.76 |
| | 4AgentAnt | 584.75 | 49.36 | -971.28 | 750.96 | -127.43 | -1005.30 | -419.93 | 149.1 | 151.85 | -457.68 | -338.21 | -164.72 |
| | 6AgentHalfCheetah | -140.96 | -302.99 | -196.46 | 1492.24 | -653.78 | -257.76 | -207.49 | -529.43 | 442.48 | -151.95 | -588.66 | -544.29 |

of five algorithms on SMAC and MPE, twelve on GRF, and ten on MAMuJoCo for community reference. All the experiments are reproducible, as we provide learning curves and complete training configurations in our code repository[3].

## 5.2 PERFORMANCE INHERITANCE OF SINGLE-AGENT RL

Empirically, we find that developing MARL algorithms based on a strong RL algorithm is a wise choice. For instance, PPO is primarily used in single-agent RL because of its better empirical performance than vanilla PG and A2C. This superiority affects the performance of their multi-agent counterparts — MAPPO and VDPPO surpass MAA2C and VDA2C in most scenarios. Another piece of evidence that corroborates this conclusion is the robustness of value iteration methods. Value iteration-based algorithms are less hyperparameter-sensitive than policy-gradient methods and more sample efficient. The multi-agent version of Q learning like IQL, VDN, and QMIX also inherits this advantage and shows robust performance in most scenarios like SMAC and MPE.

## 5.3 THE EFFECTIVENESS OF MARL ALGORITHMS

From Table 2, we find algorithms of different categorization show superiority on specific tasks that share similar task patterns.

Independent learning is effective when the central information is not necessary. While coordination among agents is essential for MARL algorithms and independent learning is theoretically suboptimal, existing work [9] has pointed out that independent learning can surpass other algorithms. In Table 2, we find that independent learning algorithms are better than their centralized critic counterparts in scenarios like `simple_spread` and `pass_and_shoot`, where agents are expected to behave similarly and central information is not necessary for policy optimization. By the same logic, without a global view, independent learning fails to solve coordination tasks such as `simple_speaker_listener` and `simple_reference`.

Centralized critic is better at learning diverse yet coordinated behaviors. In a multi-agent task, agents can take different roles, and their behaviors are expected to be role-specific [14, 34]. Centralized critic suits these tasks since local observations and global information are both well utilized. A good example is MAPPO, a representative algorithm of centralized critic that performs well on most cooperative tasks in SMAC, MPE, and GRF. HAPPO is the heterogenous version of MAPPO that achieve robust performance in MAMuJoCo. MAPPO and HAPPO are strong baselines for cooperative tasks where agents have diverse behaviors.

---

[3]https://github.com/ICLR2023Paper4242/MARLlib/tree/main/results

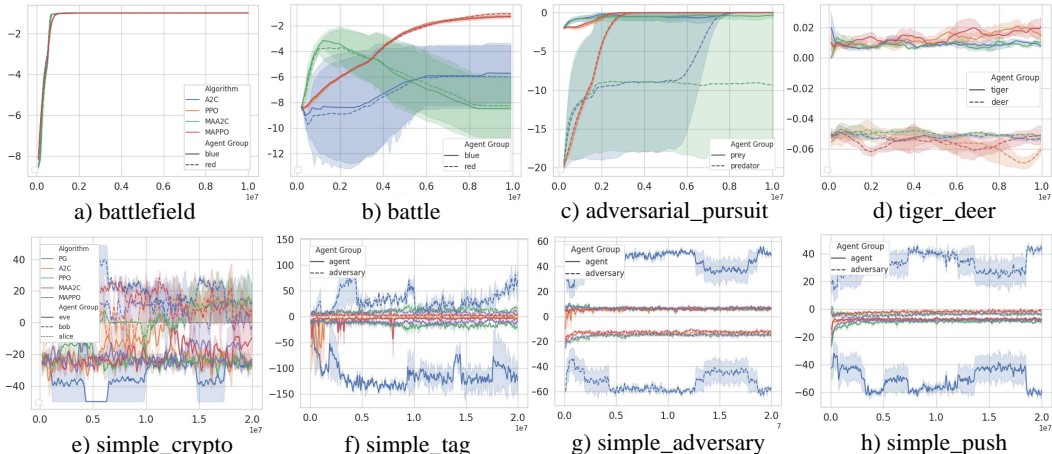

Figure 5: Reward curves of eight mixed scenarios (agents compete in group) in MAgent (a-d) and MPE (e-h). Different styles of curves stand for different agent groups. The reward curves of competing groups show a dynamic balance during the learning procedure, while the balance point depends on both algorithms and tasks.

Value decomposition dominates the popular cooperative benchmarks, except for two cases. The first case is continuous control. Well-known value decomposition algorithms like VDN and QMIX are unsuitable for continuous control tasks, and VDA2C and VDPPO are inferior compared to other algorithms. The second case is a long-term planning problem with a sparse reward function like in GRF. Empirically, the performances of value decomposition methods are significantly worse than algorithms of other categories, such as ITRPO and MAPPO. We identify two primary reasons: 1) value iteration used by VDN and QMIX prefers a dense reward function; 2) the mixer can hardly decompose a Q function close to zero. Except for these two cases, value decomposition algorithms achieve robust performance with the best sample efficiency.

## 5.4 ALGORITHM EVALUATION IN MIXED SCENARIOS

Benchmarking algorithms in mixed tasks is challenging. Agents in mixed tasks behave both cooperatively (with teammates) and competitively (to their opponents). It is hard to justify which algorithm is better based on the reward gained as the policies are always in dynamic balance: when one policy is better optimized, the performances of its opponents' policies are degraded.

Under mixed task mode, algorithms can be evaluated by the summed reward of all different policies. One policy optimization forces competitive policies to get a higher reward. Therefore, the higher the summed reward, the better the algorithms (Figure 5[a-d]). However, there are exceptions (Figure 5[e-h]). The summed reward is a constant value or around a constant value and policies quickly reach equilibrium with mirrored learning curves between competitive policies as a significant pattern. A fair and general criterion to evaluate algorithms on constant-sum tasks is still an active research direction.

## 6 CONCLUSION

In this paper, we introduced MARLlib, an integrated library that covers broad algorithms and tasks in the MARL research. MARLlib unifies diverse learning paradigms and multi-agent environment interfaces with newly proposed agent-level distributed dataflow, interface unification, and flexible parameter sharing strategies. Thousands of experiments are conducted to validate the correctness of our implementation. To summarize, MARLlib serves as a comprehensive and solid library suite for MARL algorithm training, evaluation, and comparison. We believe MARLlib can benefit large-scale multi-agent applications in the long term. Moreover, it can also benefit the MARL research community by serving as an educational portal for new researchers. At last, we strongly recommend readers to read our code and documentation where more details are served.

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

## A EFFICIENCY

We conducted experiments to show the efficiency regarding the training clock time and the memory usage of MARLlib compared to EPyMARL and On-policy baseline (official MAPPO [37]). This experiment is conducted on a local server with NVIDIA RTX A6000 GPU and AMD Ryzen Threadripper PRO 5945WX 12-Cores CPU. The testing scenario is MMM2 from SMAC [28], and the testing algorithm is MAPPO. The total consumed timesteps are $10^6$. The result can be found in Table. 3.

Table 3: Efficency and hardware usage comparison between EPyMARL, MAPPO, and MARLlib

| Framework/Benchmark | Training Clocktime (min:sec) | Memory Usage (GB) | GPU Memory Usage (MB) |
|---|---|---|---|
| EPyMARL(thread=5/10/15) | 5:29/3:14/2:24 | 8.4/12.3/15.8 | 2245/2309/2329 |
| MAPPO(thread=5/10/15) | 5:12/3:13/2:42 | 8.9/12.3/16.3 | 2157/2277/2389 |
| MARLlib(worker=5/10/15) | 3:29/2:16/1:24 | 11.2/15.4/20.4 | 5025/5327/5351 |

As a result, MARLlib is significantly more efficient than other frameworks in clock time. However, the largely increased training speed comes with relatively higher memory usage and GPU memory usage. This is partly due to Ray/RLlib's scheduling mechanism, where data is cached on GPU as long as there is still available memory. And since each agent is asked to maintain its own data buffer in MARLlib (which is a premise of tackling task modes such as competitive and mixed), the memory usage is higher than the other two.

## B EXTENSIBILITY

The extensibility of MARLlib is guaranteed by the highly abstracted architecture, which consists of five major parts: configuration, training script, algorithm, model, and environments. Each part has corresponding API, methods, or instructions explained in our documentation. Here we provide a brief description of MARLlib's extensibility gained from each part on different aspects of MARL research:

- To control the experiment deployment, such as scale up to multi-GPU parallel training, one needs to focus on the configutation part as illustrated in Sec.4 and documentation's handbook part. For instance, ray.yaml enables flexible resource allocation, such as CPU cores and numbers of available GPUs, to each worker and learner.

- To extend one algorithm to tackle more task modes, such as cooperative only to mixed scenarios, one needs to focus on the script part provided in under `marl/algo/scripts` directory. For instance, enabling an algorithm to tackle additional task modes requires reorganizing the policy mapping function in the script part.

- To enable one algorithm to tackle complex task data structures or partial observable settings, one needs to focus on `marl/models` directory. For instance, the well-known large-scale multi-agent task Neural-MMO returns an observation containing both 3D observation and 1D state.

- To develop a new algorithm, one needs to focus on the algorithm part to build new functions based on existing basic algorithm implementation under `marl/algos/core` directory and `marl/utils` directory. For instance, building a communication-based MARL algorithm requires agents to share their decision in the postprocessing function defined in `marl/algos/utils/centralized_critic.py`.

- To quickly build a baseline for newly proposed multi-agent tasks that are not already included, one needs to focus on the `envs` directory. Following the gym-like interface introduced in Sec.3.2, a new task can be easily incorporated, and all algorithms available on MARLLib can also be evaluated on this new task.

## C  TRICKS

In MARLlib, task-specific tricks are removed. Common tricks are still preserved and available to choose from. RLlib supports rich tricks in standard RL, such as gradient clip, parameter clip, value function clip, GAE($\lambda$) [29], etc. All these tricks are still available in MARLlib by modifying the hyperparameter in the configuration file. As for tricks specific to MARL, currently, there are three most commonly seen tricks in MARL.

- First is choosing a better input representation for the value function. This trick is first mentioned in MAPPO [37], where the authors find that combining local and global observation improves model performance. In MARLlib, we implement these tricks by adding additional hyperparameters in environment configuration files. For instance, to remove global observation of the critic function, simply change the `global_state_flag` to False and cut the `global_state` off in the environment return.

- Second is a suitable model size. Increasing the neuron number of a model can enlarge the model's capacity. However, a larger model size can slow down the learning speed. In MARLlib, model size can be easily controlled by changing the model configuration file. For instance, increase the number of encoder layers.

- Third is applying algorithm-specific tricks like increasing minibatch sgd-iteration number in MAPPO's training stage. These tricks are all included in MARLlib and can be adjusted by changing the parameter in the algorithms configuration.

We conducted an ablation study on adding/removing/adjusting some tricks used in MARLlib although there is no comparable result reported before. We test one trick on MAPPO and two on QMIX [26]. The testing bed is SMAC scenario MMM2. Different groups use different tricks combination, indicated by ✓.

Table 4: Input representation trick on MAPPO

|         | local observation | global state | opponent action | reward |
|---------|-------------------|--------------|-----------------|--------|
| Group 1 | ✓                 |              |                 | 15.74  |
| Group 2 | ✓                 | ✓            |                 | 16.13  |
| Group 3 | ✓                 | ✓            | ✓               | 16.86  |

Table 5: Reward normalization trick on QMIX with different model size

|         | layer | dimension   | reward norm | reward |
|---------|-------|-------------|-------------|--------|
| Group 1 | 1     | 128         |             | 5.68   |
| Group 2 | 1     | 128         | ✓           | 18.34  |
| Group 3 | 2     | 128\|64     |             | 4.71   |
| Group 4 | 2     | 128\|64     | ✓           | 16.23  |
| Group 5 | 3     | 256\|128\|64 |            | 3.76   |
| Group 6 | 3     | 256\|128\|64 | ✓          | 15.98  |

As a result, we find the input representation has a minor effect on the final performance of the centralized critic-based method MAPPO. At the same time, tricks like reward normalization have a significant impact on the value iteration-based method QMIX.

## D RUNNING OPTIONS

MARLlib provides three options of building and running MARLlib, including standard, docker-based, and website-based.

- Standard way is best for learning the workflow of MARLlib and developing new algorithms on a local machine.
- Docker-based way is proposed for fast deployment and testing the framework's utility.
- Website-based way is most suitable for the visualization of training results and easy configuration manipulation.

## E DOCUMENTATION

MARLlib is proposed with documentation affiliated. The documentation consists of four major parts.

- The first part is the MARLlib handbook, which contains the installation of MARLlib and the usage of ten incorporated environments[4].
- The second part is guidance for new MARL researchers to navigate from RL to MARL, with a collective survey of MARL[5].
- The third part is the algorithm's documentation, with five self-contained algorithm families, discussing the internal relationship within different algorithms and how they are organized in MARLlib[6].
- The fourth part is the resources of MARL, including an awesome list of algorithm papers and existing MARL benchmarks[7].

This documentation provides MARLlib users with a comprehensive understanding of MARL's basic idea and representative algorithms. We recommend MARLlib users have basic ideas of DL and RL. Still, when reading the documentation, users can easily find the background knowledge and related information with a linked URL or brief introduction.

## F MAPPO BENCHMARK, EPYMARL, AND MARLLIB

To further ensure the MARLlib algorithm implementation's trustworthiness and correctness, we conduct extra experiments that compare the performance (reward gain) of MARLlib and the official MAPPO implementation[37] on widely-used SMAC benchmark [28] as the testing bed. Here we also include EPyMARL's 20 million timesteps performance for reference.

| Framework/Benchmark | 2s_vs_1sc | 3s5z | MMM2 | 3s_vs_5z |
|---|---|---|---|---|
| EPyMARL | 20.20 | 18.71 | 16.67 | 17.62 |
| MAPPO | 20.21 | 18.34 | 18.20 | 13.78 |
| MARLlib | 20.21 | 19.52 | 16.86 | 17.24 |

As shown in the table above, the model performance among all three benchmarks is similar, which proves that the implementation in MARLlib is trustworthy.

---

[4]https://iclr2023marllib.readthedocs.io/en/latest/handbook/intro.html

[5]https://iclr2023marllib.readthedocs.io/en/latest/intro_marl/rl.html

[6]https://iclr2023marllib.readthedocs.io/en/latest/algorithm/jointQ_family.html

[7]https://iclr2023marllib.readthedocs.io/en/latest/resources/awesome.html

