# OpenReview forum: "MARLlib: Extending RLlib for Multi-agent Reinforcement Learning"
_ICLR.cc/2023/Conference — Submitted to ICLR 2023_

### Official Review · Reviewer_o2ZC · 2022-10-23

**Confidence:** 3
**Correctness:** 3
**Technical Novelty And Significance:** 2
**Empirical Novelty And Significance:** 1
**Recommendation:** 5

**Clarity, Quality, Novelty And Reproducibility:**

- Clarity: The paper is readable. Not exceptionally clear but the text was not confusing or hard to follow.
- Quality: Effectively address the problem of a unified algorithm and environment interface for MARL (at least this definition of unified).
- Originality: Incrementally extends efforts at general MARL interfaces by explicitly supporting diverse task settings.
- Reproducibility: Looks very good. Code repository is included and all results are hosted there as well.

**Strength And Weaknesses:**

## Strengths

The main strengths of the paper:
1. Algorithm versatility. The parameter sharing implementation and data flow structure supports a meaningful subset of MARL implementations.
2. Environment breadth. Including cooperative, competitive, and mixed scenarios facilitates a wide variety of research. This may also encourage further development of algorithms that perform well across all contexts and deeper analysis of how task vs environment structure influence algorithm performance.


## Weaknesses

1. No evidence on runtime performance. How fast and scalable is MARLlib compared to alternatives? Can researchers quickly run experiments given typical hardware? How does the throughput vary with number of agents? How close are the algorithm implementations to their competitors in terms of runtime efficiency? Clarifying runtime characteristics would strengthen the case that MARLlib is a useful addition to the tools available for running MARL experiments.
2. Limited evidence compared to existing algorithms. The only comparison made is to EPyMARL, with no reference baselines provided for other environment algorithms.
3. Lack of baseline comparisons. The configurations used remove many "tricks" applied in other algorithms. It is hard to gauge if the implementations are correct due to the changes needed to fit the config files. It would help to include reference to previously reported performance numbers for comparison.


## Feedback & Questions

Table 2 would benefit from reporting baseline performance results from other implementations for comparison.

Runtime performance merits a discussion. Specifically addressing scaling in terms of memory and runtime, when varying number of agents, and so on. This would benefit from comparisons to existing methods.

Some questions:
- Can MARLlib support agents that have different lifespans (not around entire task)? That is, tasks where agents "die" during the task.
- What are the "virtual" vs "physical" policies mentioned in section 3.3? Does physical mean the data structures used?
- How can developers/users add "tricks" to their algorithms that are not already provided by the configuration?

**Summary Of The Paper:**

The paper introduces a library - MARLlib - to run multi-agent reinforcement learning experiments with different environments, task settings (cooperative, competitive, or mixed), and algorithms. The core contribution is defining how observations, predictions, and other data are shared among agents that are otherwise distributed during training. The structure defined is flexible enough to support existing MARL algorithms that vary in using centralized critics or mixing functions for value decomposition and supports environments that use synchronous or asynchronous updates. In practice the interface modifies the standard gym definition to accommodate per-agent data and uses RLlib to provide parameter sharing functionality. Experiments include a comparison with EPyMARL algorithm implementations in SMAC and include results on several other MARL benchmarks (MPE, GRF, MAMuJoCo).

**Summary Of The Review:**

The contributions of this paper come from unifying existing approaches and the novelty of the formulation of that unification. From a technical standpoint the novelty is a distributed dataflow formulation to enable distributed agent learning through decoupling agents from each other during optimization. This is a useful insight, but it is not clear it unifies all the existing MARL approaches nor does the paper explore what new algorithms may be facilitated. The main contribution is the experiment platform provided, more than the insights facilitated by its conceptual foundation.

The empirical results lack baselines in most cases. Thus, it is hard to tell if the implementations are "correct" relative to previously published results. The results do not claim any novelty and are intended to demonstrate comparability instead. Without proper baselines it is only possible to assess that EPyMARL and MARLlib offer roughly similar agent training convergence in many, but not all, cases (ex: IQL and MAA2C in 3s_vs_5z do substantially worse). No empirical results offer insight into the runtime performance of the platform, which would be important if researchers are to adopt the tool in lieu of other more targeted environments.

Taken together, the paper offers a potentially useful platform for others to consider. The lack of baselines makes it difficult to gauge whether the implementations being provided are close to other options. Without runtime or other performance data it is hard to tell if this framework is useful for others to adopt instead of existing options.

---

> ### Author Response · Authors · 2022-11-16
> **Authors Response to Reviewer o2ZC Part 2**
>
>
> > Q5: What are the "virtual" vs. "physical" policies mentioned in section 3.3? Does physical mean the data structures used?
>
> A5: Thank you for the valuable question. The virtual policy is the agent-level policy. A group of agents may share one physical policy to get a copy of that, meaning the physical policy is task-level. For instance, when N agents fully share the model parameters, there are N virtual policies (each agent takes one) and only one physical policy (cause it is shared across N agents). The use of "virtual" and "physical" is inspired by the concept in operation systems such as virtual memory and physical memory to help MARLlib users catch their meanings of them and better understand the policy mapping function.
>
>
>
> > Q6: How can developers/users add "tricks" to their algorithms that are not already provided by the configuration?
>
> A6: Please refer to A3.
>
>
>
> > Q7: Performance difference between EPyMARL and MARLlib in some cases
>
> A7: Thank you for raising the question. As summarized in Sec 5.1, compared to EPyMARL, MARLlib attains similar results on 63% of them (the reward difference is less than 1.0), and achieves superior results on 25% of them. Specific to 3s\_vs\_5z, among eight algorithms, four are much better than EPyMARL, two are similar, and two are significantly worse. There are two major causes of the performance difference:
>
> - First is the tricks used. For instance, EPyMARL uses reward standardization and manually selects TD(n) for computing target value, while we use standard GAE($\lambda$) instead, which is more widely adopted.
> - Second is the different way of algorithm implementation. For instance, EPyMARL implements all its on-policy algorithms (e.g., MAPPO, MAA2C) based on MADDPG. For instance, the model of MAPPO in EPyMARL is equipped with a target actor/critic network to produce target value and use it to optimize the policy. However, from standard RL's perspective, the target network is only useful in the off-policy RL training. On-policy RL, such as PPO, is a regression problem that no target network is needed. Compared to EPyMARL, MARLlib implements algorithms based on widely acknowledged on-policy implementation such as Tianshou, RLlib, and Spinning Up. The official implementation of MAPPO also adopts this way of implementing its algorithm. Therefore, tricks use and implementation differences are the major causes of the performance difference between EPyMARL and MARLlib.

---

> ### Author Response · Authors · 2022-11-16
> **Authors Response to Reviewer o2ZC Part 1**
>
>
> Thanks for your comments. We address your major concerns one by one.
>
> > Q1: No evidence of runtime performance. How fast and scalable is MARLlib compared to alternatives? Can researchers quickly run experiments given typical hardware? How does the throughput vary with number of agents? How close are the algorithm implementations to their competitors in terms of runtime efficiency? Clarifying runtime characteristics would strengthen the case that MARLlib is a useful addition to the tools available for running MARL experiments.
>
>
> A1: Thank you very much for your valuable comments and guidance on improving the paper. We conducted extra experiments to show the efficiency regarding the training clock time and the memory usage of MARLlib compared to EPyMARL and On-policy baseline (official MAPPO). We have incorporated this part of the evaluation and discussion in the paper's Appendix A.
>
>
> > Q2: Limited evidence compared to existing algorithms. The only comparison made is to EPyMARL, with no reference baselines provided for other environment algorithms.
>
> A2: To the best of our knowledge, EPyMARL is the only existing benchmark that unifies diverse MARL algorithms and tasks in one framework, thus is close to MARLlib and suitable to compare with. We cannot provide the result of MPE, GRF, MAMuJoCo from EPyMARL because:
> - we use MPE maintained by PettingZoo. In contrast, EPyMARL uses its own version, which leads to an unfair comparison.
> - GRF & MAMuJoCo are not supported by EPyMARL technically because of their task settings, e.g., continuous action space and 3D observation, while EPyMARL only supports discrete action and 1D input.
>
> Other than the benchmark level, comparison can only be conducted at the algorithm level, which requires the official implementation of each algorithm listed. Different tricks used in such implementation will significantly impact the performance comparison. Therefore, we can not ensure the evaluation result is fair between MARLlib and one single performance. However, to further ensure the MARLlib algorithm implementation's trustworthiness and correctness, we conduct extra experiments that compare the performance (reward gain) of MARLlib and the official MAPPO benchmark. Here we also include EPyMARL's performance for reference (20 million steps). We have incorporated this part of the evaluation and discussion in the paper's Appendix F.
>
>
>
> > Q3:  Lack of baseline comparisons. The configurations used remove many "tricks" applied in other algorithms. It is hard to gauge if the implementations are correct due to the changes needed to fit the config files. It would help to include reference to previously reported performance numbers for comparison
>
> A3:  Thank you for pointing it out. We have incorporated this part of the evaluation and discussion in the paper's Appendix C.
>
>
>
> > Q4: Can MARLlib support agents with different lifespans (not around the entire task)? That is tasks where agents "die" during the task.
>
> A4: Yes. Tasks such as MetaDrive and Hanabi are exactly this kind: agents can die early (no cards in hand or one agent reaches the individual target location), while other agents keep interacting with the environment.

---

### Official Review · Reviewer_RpXo · 2022-10-23

**Confidence:** 4
**Correctness:** 2
**Technical Novelty And Significance:** 2
**Empirical Novelty And Significance:** 2
**Recommendation:** 3

**Clarity, Quality, Novelty And Reproducibility:**

The paper is in general well-written and easy to follow.

My biggest concern is regarding the reproducibility. In fact, I spent sometime to try to understand the new library based on the provided link to the source code and online documentation. The library does not seem to work well. I experienced a lot of errors while trying to test some multi-agent reinforcement learning algorithms supported by the library.

Below I list some errors I found during my trial runs:

Change pip install gym==1.21.0 to pip install gym==0.21.0 in the readme file.

The icecream library is not included in the pip list but is used in centralized_critic_hetero.py and happo.py

The command on running an example on SMAC --algo_config=MAPPO should be --algo_config=mappo

When we do python add_patch.py -y -p, Pommerman must be installed since it is referenced somewhere in the library.

The given example in the readme is not working. I tried other examples, none of them seem to work properly.

Additionally, I found that many algorithms claimed to be supported by the new library were already implemented/supported by RLlib and Ray. For the newly implemented algorithms in the library, they do not seem work without errors.

**Strength And Weaknesses:**

Strength:
With the increasing popularity of multi-agent reinforcement learning, it is important to develop a reliable and high-quality multi-agent reinforcement learning platform to foster fruitful future research in this booming field. This paper introduced such a platform with some interesting system design ideas, such as agent-level distributed dataflow that unifies diverse learning paradigms. Experiment results show that this platform provides high-quality implementation of many existing multi-agent reinforcement learning algorithms and can be very useful for researchers who are interested in developing and using multi-agent reinforcement learning technologies.

Weakness:
Many libraries have been developed to support single-agent reinforcement learning. It is not clear why the authors chose to use RLlib as the basis for developing their new multi-agent reinforcement learning platform. Is it possible for the multi-agent platform to support multiple existing reinforcement learning libraries? This might be helpful to satisfy researchers' diverse preferences over reinforcement learning libraries.

To determine the overall usefulness of the newly developed multi-agent platform, it might be important to understand which computing platform and infrastructure it supports. Furthermore, it is also interesting and important to know whether the new platform supports distributed deployment for many practical applications. Can the platform support scalable learning through large-scale parallel processing? Is the platform compatible with (or support) any publicly accessible parallel computing services and facilities?

For any platform to be successful, it needs to meet certain efficiency requirements. While the authors evaluated the effectiveness of several multi-agent reinforcement learning algorithms that are supported by the new platform, in terms of computation time and computation resources required, it is not clear whether the new platform is more competitive than any existing platforms.

Another major concern is the user experience. I understand that the new platform supports some ease-of-use features. For example, users can configure several aspects of their multi-agent reinforcement learning pipeline. In line with this feature, I was wondering whether the platform also allows users to configure certain details regarding experiment deployment, experiment progress monitoring, result collection and reporting. Perhaps a case study can also be carried out to demonstrate how the new platform can help researchers develop and evaluate their new multi-agent reinforcement learning algorithms. Additionally, the extensibility of the new platform may need to be examined in more depth.

In line with the above, the robustness of the new platform implementation may also need to be verified and reported in the paper. For example, it might be interesting to find out how likely the new platform may run into any errors (for example errors due to instability of numerical calculation) while conducting a large-scale experiment.

According to my understanding, the new platform does not support some algorithm-specific tricks. While this is understandable to a certain extent, it remains unclear about (1) whether users can easily implement and support any tricks while conducting experiments using the new platform and (2) due to the absence of algorithm-specific tricks, how big the performance difference could be between the results published by the respective algorithm inventors and the results obtained from using the new platform. Moreover, the paper only compared the performance of several algorithms by using two alternative multi-agent reinforcement learning platforms. As only four random seeds are used and the performance differences cannot be verified statistically, it remains unclear whether the new platform indeed provides high-quality implementations of all the multi-agent reinforcement learning algorithms it supports.

Perhaps the authors can comment more about documentation support for the new platform. The usefulness of any software platform depends on its documentation support. The authors may want to discuss how their documentation can help researchers quickly learn their platform. What kind of background knowledge researchers must have in order to understand the documentation? This question can be clarified in the paper too.

**Summary Of The Paper:**

This paper introduced a new development of a multi-agent reinforcement learning platform based on RLlib to support cutting-edge research on multi-agent reinforcement learning technologies. This paper introduced several key platform design ideas that help to build a flexible and general platform for developing and experimenting multi-agent reinforcement learning algorithms.

**Summary Of The Review:**

The paper considered an important problem of building a flexible and extensible platform for multi-agent reinforcement learning. While building such a platform clearly has significant research values, I have major concerns regarding the reproducibility, the quality of the code and the correctness/clarity of the online documentation.

---

> ### Author Response · Authors · 2022-11-16
> **Authors Response to Reviewer RpXo Part 3**
>
> > Q5: In line with the above, the robustness of the new platform implementation may also need to be verified and reported in the paper. For example, it might be interesting to find out how likely the new platform may run into any errors (for example, errors due to instability of numerical calculation) while conducting a large-scale experiment.
>
> A5: Thank you very much for the comment. We have yet to meet any running time error after conducting over 1000 experiments on MARLlib, both locally and on cloud servers. As MARLlib is mainly proposed for research use and due to our hardware limitation, we can not conduct a large-scale experiment to evaluate the stability of MARLlib on a big cluster with hundreds of computing nodes. In the future, we will add a troubleshooting section based on users' bug report and issue raised.
>
>
>
> > Q6.1: According to my understanding, the new platform does not support some algorithm-specific tricks. While this is understandable to a certain extent, it remains unclear about (1) whether users can quickly implement and support any tricks while conducting experiments using the new platform and (2) due to the absence of algorithm-specific tricks, how big the performance difference could be between the results published by the respective algorithm inventors and the results obtained from using the new platform.
>
> A6.1: Thank you very much for raising the question.
> RLlib supports rich tricks in standard RL, such as gradient clip, parameter clip, value function clip, GAE(\lambda), etc. All these tricks are still available in MARLlib by modifying the hyperparameter in the configuration file.
> For tricks specific to MARL, we have incorporated them into the discussion in the paper's Appendix C.
>
>
>
>
> > Q6.2: Moreover, the paper only compared the performance of several algorithms by using two alternative multi-agent reinforcement learning platforms. As only four random seeds are used, and the performance differences cannot be verified statistically, it remains unclear whether the new platform indeed provides high-quality implementations of all the multi-agent reinforcement learning algorithms it supports.
>
> A6.2:Thank you very much for the valuable question. We conduct four seeds per algorithm-scenario pair due to the large number of experiments we need to conduct (over 1000). In addition, according to the existing benchmark EPyMARL where one performance is averaged over five random seeds, the variance scope is acceptable. Therefore, we conduct the experiment for extra two to four seeds to study the impact on the final result.
>
> | Performance(reward) | 2s\_vs\_1sc |3s5z  | MMM2 | 3s\_vs\_5z |
> | ----------- | ----------- | ----------- | ----------- | ----------- |
> | MAPPO (4/6/8 seeds average) | 20.21/20.23(+0.02)/20.22(+0.01)| 19.52/19.27(-0.25)/19.43(-0.09) | 16.86/17.01(+0.15)/16.92(+0.06) |17.24/18.11(+0.87)/18.44(+1.20)| |
>
> From the table above, we can see that with more seeds used in the experiment, the model performance varies in a small range. We will conduct more experiments to ensure the correctness of the final results.
>
> > Q7: Perhaps the authors can comment more about documentation support for the new platform. The usefulness of any software platform depends on its documentation support. The authors may want to discuss how their documentation can help researchers quickly learn their platform. What kind of background knowledge researchers must have in order to understand the documentation? This question can be clarified in the paper too.
>
> A7: Thanks for your suggestion. There are four major parts of our documentation, and we have incorporated this part of the description in the paper's Appendix E.
>
> > Q8: Issues of MARLlib
>
> A8: Thank you very much for the helpful comments on our user instructions. We apologize for causing the confusion. We fix the error in our instructions, including
>
> - replace gym\==1.21.0 with gym\==0.21.0
> - include icecream package installation
> - fix the typo error in the running example
> - clarify that only with pommerman installed can add `-p` flag in `add_patch.py -y'
> - all the algorithms are tested with no bug reported
>
> Furthermore, to ensure a smooth and convenient user experience, we provide a Docker-based option for MARLlib users, with a dockerfile in the github repo and a docker image in docker hub for running the code. The instruction of docker usage can be found [here](https://github.com/ICLR2023Paper4242/MARLlib#docker).
>
> Command for downloading the ready-to-run image: `docker pull iclr2023paper4242/marllib:1.0`.

---

> ### Author Response · Authors · 2022-11-16
> **Authors Response to Reviewer RpXo Part 2**
>
>
>
> > Q3: For any platform to be successful, it needs to meet certain efficiency requirements. While the authors evaluated the effectiveness of several multi-agent reinforcement learning algorithms that are supported by the new platform in terms of computation time and computation resources required, it is not clear whether the new platform is more competitive than any existing platforms.
>
> A3: Thank you very much for the valuable suggestion. We conducted extra experiments to show the efficiency regarding the training clock time and the memory usage of MARLlib compared to EPyMARL and On-policy baseline (official MAPPO). We have incorporated this part of the evaluation and discussion in the paper's Appendix A.
>
>
>
> > Q4.1: Another major concern is the user experience. I understand that the new platform supports some ease-of-use features. For example, users can configure several aspects of their multi-agent reinforcement learning pipeline. In line with this feature, I was wondering whether the platform also allows users to configure certain details regarding experiment deployment, experiment progress monitoring, result collection and reporting.
>
> A4.1: Thank you very much for the comment. MARLlib supports customized experiment deployment, experiment progress monitoring, and result collection and reporting in different ways. We can customize part of them through configuration, such as the parallel sampling worker number in the experiment deployment or early stop/trigger setting when a certain condition is met in experiment progress monitoring. While the rest of them, such as result reporting, can be modified through code-level function inheritance. For instance, in order to customize the result collected or reported, one needs to inherit the `DefaultCallbacks` class and collect needed metrics/results in different stages of training.
>
>
> > Q4.2: Perhaps a case study can also be carried out to demonstrate how the new platform can help researchers develop and evaluate their new multi-agent reinforcement learning algorithms. Additionally, the extensibility of the new platform may need to be examined in more depth.
>
> A4.2: Thank you for the precious suggestion. We fully illustrate the extensibility of MARLlib in response to R2 A4 from five aspects covering algorithm, environment, model, configuration, and deployment, where quickly building new algorithms is one of them. Here we raise an example of how to build a communication-based MARL algorithm as an example. Communication-based agent model should have three modules: policy function, value function, and message function. The only difference between communication-based PPO (here we take PPO as the base method, refer to it as comm-PPO) and MAPPO agent in MARLlib is the additional communication process: agents produce a message via message function and share it with others. To implement a comm-PPO in MARLlib, we can reuse MAPPO's learning pipeline and modules with some tweaks in several places to transform MAPPO to comm-PPO.
> - The first place is [model part](https://github.com/ICLR2023Paper4242/MARLlib/tree/main/marl/models), where we can add a message function on currently available models.
> - The second place is [centralized critic function](https://github.com/ICLR2023Paper4242/MARLlib/blob/main/marl/algos/utils/centralized_critic.py), where the agent shares the message with others and records the message others send.
> - The third place is [loss function](https://github.com/ICLR2023Paper4242/MARLlib/blob/main/marl/algos/core/CC/mappo.py), where the policy gradient update has to be replaced by a communication-based one based on PPO loss.

---

> ### Author Response · Authors · 2022-11-16
> **Authors Response to Reviewer RpXo Part 1**
>
>
> Thank you very much for your helpful comments and precious suggestions, and we would like to address them as follows.
>
>
> > Q1.1: Many libraries have been developed to support single-agent reinforcement learning. It is not clear why the authors chose to use RLlib as the basis for developing their new multi-agent reinforcement learning platform.
>
> A1.1: RLlib is based on Ray, a job scheduling framework for general machine learning, enabling highly distributed RL. At the same time, RLlib maintains unified and straightforward APIs for RL training and can cover most research topics in RL, such as offline RL and multi-agent RL on the execution level. Compared to other single-agent reinforcement learning frameworks like Spinning Up or Tianshou, we prefer RLlib because of its rich support for all kinds of RL learning paradigms, its trustful implementation of algorithms, and its scalability for all kinds of applications.
>
> Standing on the shoulder of RLlib, algorithms in MARLlib can fully utilize the advantages that Ray/RLlib can offer:
>
> 1.  Excellent efficiency and flexibility of RL (e.g., distributed sampling/training)
> 2.  Easy-to-use plug-in tricks (e.g., GAE)
> 3. Reproducible results (e.g., complete training configuration record)
> 4. Execution-level support for multi-agent RL
> 5. Trustworthy RL algorithms implementation
>
> Due to the reasons above, we choose to use RLlib as our basis.
>
>
>
> > Q1.2 Is it possible for the multi-agent platform to support multiple existing reinforcement learning libraries? This might be helpful in satisfying researchers' diverse preferences over reinforcement learning libraries.
>
> A1.2: Thank you very much for the enlightening question. We believe it is technically challenging to make one MARL framework support multiple RL frameworks because the API of popular RL frameworks is exposed for direct use or already at the application level. The key contribution of an RL framework is its unique learning pipeline abstraction. However, if we choose to build a MARL algorithm based on one framework, the implementation must be deeply coupled with this abstraction. Therefore,  It is nearly impossible to distill various abstract APIs from different RL frameworks and unify them into one. Besides, most popular RL frameworks, such as Tianshou and RLlib are under active development, making a MARL framework very unstable if built upon them.
>
>
>
> > Q2: To determine the overall usefulness of the newly developed multi-agent platform, it might be important to understand which computing platform and infrastructure it supports. Furthermore, it is also interesting and important to know whether the new platform supports distributed deployment for many practical applications. Can the platform support scalable learning through large-scale parallel processing? Is the platform compatible with (or support) any publicly accessible parallel computing services and facilities?
>
> A2: Thank you very much for the helpful suggestion. As illustrated in A1.1, MARLlib can fully utilize the feature of Ray/RLlib. The infrastructure-level support is the same as Ray/RLlib. For instance, by setting the `num_workers` configuration parameter, algorithms like A3C or IMPALA can run on 100s of CPUs/nodes, thus parallelizing and speeding up learning. Compared to existing MARL frameworks/benchmarks like MAPPO and EPyMARL, which can only be deployed on a single-node machine, MARLlib offers greater flexibility for MARLlib users to deploy on diverse computing platforms and infrastructure. Still, we want to emphasize that the purpose and main contribution of MARLlib is not large-scale deployment ability but providing a unified platform to benefit research where diverse algorithms and environments can be put together and share the same learning pipeline.

---

### Official Review · Reviewer_EKiW · 2022-10-25

**Confidence:** 3
**Correctness:** 3
**Technical Novelty And Significance:** 3
**Empirical Novelty And Significance:** 3
**Recommendation:** 5

**Clarity, Quality, Novelty And Reproducibility:**

Clarity: The paper is easy to follow.
Novelty: The paper is somewhat novel, but the novelty should be further emphasized
Reproducibility: the well-implemented code is provided, so it should be reproducible.

**Strength And Weaknesses:**

**Strength**
- This work implements a relatively large number of MARL algorithms on multiple environments. This is useful for researchers to compare new algorithms to existing baselines.
- Extensive evaluation is conducted for the proposed framework
- The proposed framework integrates a large number of environments

**Weaknesses and Questions**
- As a unified framework to support the future development of MARL, the correctness of algorithms is important. For example, in Table 2, for SMAC 3s_vs_5z and IPPO algorithm, it seems the performance difference between your framework and EPyMARL is relatively big. Why is there such a difference?
- As a unified frame, how does your framework compare to other frameworks in terms of performance? It would be good to have some evaluation results (e.g. training clock time, memory usage, etc) both in the single machine case and distributed case.
- The original RLlib also has some support for MARL, how does your implementation differ from it? Providing more discussion in terms of your contributions and difference can help the reader better understand the framework.
- Extensibility is also important for a unified framework, how does your framework support future development of new MARL algorithms? Do you provide API, methods, or instructions that can make future development MARL algorithm easy?



**Summary Of The Paper:**

This work proposes a unified coding framework for MARL based on RLlib and Ray. The frame support a relatively large number of different algorithms (of different types), and a relatively large number of MARL environments.

**Summary Of The Review:**

Overall, this work can be useful for the MARL community, but the contribution should be clarified and concerns mentioned above should be considered.

---

> ### Author Response · Authors · 2022-11-16
> **Authors Response to Reviewer EKiW Part 2**
>
> > Q3: The original RLlib also has some support for MARL, how does your implementation differ from it? Providing more discussion regarding your contributions and difference can help the reader better understand the framework.
>
> A3: Thank you for pointing it out. Please allow us to answer it in the most respectful manner.
>
> RLlib's support on multi-agent focuses on the execution level, not the tool level, which is to say RLlib provides a potential solution to execute a MARL algorithm but has not reached the level where one can pick one MARL algorithm out of the box and directly applied to any environments/tasks. Building a single algorithm is easy, but building a unified framework for diverse MARL algorithms is hard. For instance, extending PPO to IPPO in RLlib is simple without any code-wise adjustment. The challenge is building a benchmark where different algorithms with different learning pipelines (e.g., IPPO, MAPPO, and VDPPO) can share most of the modules, including sampling, processing, model architecture, and training can work regardless of the multi-agent task-wise change,  which is what MARLlib achieved. Different algorithms can then be put into fair comparison, and new algorithms can be developed once and evaluated on all multi-agent tasks. In addition, functions and APIs such as postprocessing and policy mapping are fully utilized in MARLlib and provide a new perspective for MARL researchers and developers to extend and fast evaluate their idea based on existing MARL algorithms. Last but not least, MARLlib provides detailed documentation on how to understand each algorithm from both theoretical and implementation perspectives. In a sentence, MARLlib is a big extension of RLlib that fully utilize the advance of Ray/RLlib and provide rich algorithms, and environment supports for MARL research and application.
>
> > Q4: Extensibility is also important for a unified framework, how does your framework support the future development of new MARL algorithms? Do you provide API, methods, or instructions that can make future development MARL algorithm easy?
>
> A4: Thank you very much for the enlightening question. The extensibility of MARLlib is guaranteed by the highly abstracted architecture, which consists of five major parts: configuration, training script, algorithm, model, and environments. Each part has corresponding API, methods, or instructions explained in our documentation. We have incorporated this part of the discussion in the paper's Appendix B.

---

> ### Author Response · Authors · 2022-11-16
> **Authors Response to Reviewer EKiW Part 1**
>
> Thanks for your comments. We address your major concerns one by one.
>
> > Q1: As a unified framework to support the future development of MARL, the correctness of algorithms is important. For example, in Table 2, for SMAC 3s\_vs\_5z and IPPO algorithm, it seems the performance difference between your framework and EPyMARL is relatively big. Why is there such a difference?
>
> A1: Thank you for raising the question. As summarized in Sec 5.1, compared to EPyMARL, MARLlib attains similar results on 63% of them (the reward difference is less than 1.0) and achieves superior results on 25% of them. Specific to 3s\_vs\_5z, among eight algorithms, four are much better than EPyMARL, two are similar, and two are significantly worse. There are two major causes of the performance difference:
>
> - First is the tricks used. For instance, EPyMARL uses reward standardization and manually selects TD(n) for computing target value, while we use standard GAE($\lambda$) instead, which is more widely adopted.
> - Second is the different way of algorithm implementation. For instance, EPyMARL implements all its on-policy algorithms (e.g., MAPPO, MAA2C) based on MADDPG. For instance, the model of MAPPO in EPyMARL is equipped with a target actor/critic network to produce target value and use it to optimize the policy. However, from standard RL's perspective, the target network is only useful in the off-policy RL training. On-policy RL, such as PPO, is a regression problem that no target network is needed. Unlike EPyMARL, MARLlib implements algorithms based on widely acknowledged on-policy implementation, such as Tianshou, RLlib, and Spinning Up. The official implementation of MAPPO also adopts this way of implementing its algorithm. Therefore, tricks use and implementation differences are the major causes of the performance difference between EPyMARL and MARLlib.
>
> However, the difference is insignificant in most scenarios, as we showed in Table 2.
>
>
>
> > Q2: As a unified frame, how does your framework compare to other frameworks in terms of performance? It would be good to have some evaluation results (e.g., training clock time, memory usage, etc.) both in the single-machine case and distributed case.
>
> A2: Thank you very much for the valuable suggestion. We conducted an extra experiment to show the efficiency regarding the training clock time and the memory usage of MARLlib compared to EPyMARL and On-policy baseline (official MAPPO) on SMAC map MMM2. We have incorporated this part of the evaluation and discussion in the paper's Appendix A.

---

### Official Review · Reviewer_Fhyv · 2022-10-26

**Confidence:** 3
**Clarity, Quality, Novelty And Reproducibility:** Original work.
**Correctness:** 3
**Technical Novelty And Significance:** 2
**Empirical Novelty And Significance:** 2
**Recommendation:** 5

**Strength And Weaknesses:**

Strengths:
1.	An integrated library suite for MARL algorithms, Multi-Agent RLlib (MARLlib), is introduced.
2.	MARLlib unifies diverse environment interfaces with a newly proposed Gym-style interface.
3.	Open source codebase
Weaknesses:
1.	For example, in the 3s_vs_5z map of SMAC, why the performance conducted in MARLib has an obvious different from EPyMARL?
2.	For other environments (MPE, GRF, MAMuJoCo), why the performances of MARLlib are only included?
3.	Author may clearly illustrate the benefits obtained from agent-level distributed dataflow, a unified agent-environment interface, and effective policy mapping in terms of implementations or experiments.
4.	Some statements, like “value iteration used by VDN and QMIX prefers a dense reward function”, lack solid explanations.

**Summary Of The Paper:**

In this paper, authors propose a comprehensive MARL algorithm library (MARLlib) for solving multi-agent problems. MARLlib manages to unify tens of algorithms, including different types of independent learning, centralized critic, and value decomposition methods. And MARLlib goes beyond current work by integrating diverse environment interfaces and providing flexible parameter sharing strategies

**Summary Of The Review:**

In this paper, authors propose a comprehensive MARL algorithm library (MARLlib) for solving multi-agent problems. MARLlib manages to unify tens of algorithms, including different types of independent learning, centralized critic, and value decomposition methods. And MARLlib goes beyond current work by integrating diverse environment interfaces and providing flexible parameter sharing strategies
Strengths:
1.	An integrated library suite for MARL algorithms, Multi-Agent RLlib (MARLlib), is introduced.
2.	MARLlib unifies diverse environment interfaces with a newly proposed Gym-style interface.
3.	Open source codebase
Weaknesses:
1.	For example, in the 3s_vs_5z map of SMAC, why the performance conducted in MARLib has an obvious different from EPyMARL?
2.	For other environments (MPE, GRF, MAMuJoCo), why the performances of MARLlib are only included?
3.	Author may clearly illustrate the benefits obtained from agent-level distributed dataflow, a unified agent-environment interface, and effective policy mapping in terms of implementations or experiments.
4.	Some statements, like “value iteration used by VDN and QMIX prefers a dense reward function”, lack solid explanations.

---

> ### Author Response · Authors · 2022-11-16
> **Authors Response to Reviewer Fhyv**
>
> Thanks for your comments. We address your major concerns one by one.
>
>
> >  Q1: For example, in the 3s\_vs\_5z map of SMAC, why the performance conducted in MARLib has an obvious different from EPyMARL.
>
> A1: Thank you for raising the question. As summarized in Sec 5.1, compared to EPyMARL, MARLlib attains similar results on 63% of them (the reward difference is less than 1.0), and achieves superior results on 25% of them. Specific to 3s\_vs\_5z, among eight algorithms, four are much better than EPyMARL, two are similar, and two are significantly worse. There are two major causes of the performance difference:
>
> - First is the tricks used. For instance, EPyMARL uses reward standardization and manually selects TD(n) for computing target value, while we use standard GAE($\lambda$) instead, which is more widely adopted.
> - Second is the different way of algorithm implementation. For instance, EPyMARL implements all its on-policy algorithms (e.g., MAPPO, MAA2C) based on MADDPG. For instance, the model of MAPPO in EPyMARL is equipped with a target actor/critic network to produce target value and use it to optimize the policy. However, from standard RL's perspective, the target network is only useful in the off-policy RL training. On-policy RL, such as PPO, is a regression problem that no target network is needed. Unlike EPyMARL, MARLlib implements algorithms based on widely acknowledged on-policy implementation, such as Tianshou, RLlib, and Spinning Up. The official implementation of MAPPO also adopts this way of implementing its algorithm. Therefore, tricks use and implementation differences are the major causes of the performance difference between EPyMARL and MARLlib.
>
> However, the difference is insignificant in most scenarios, as we showed in Table 2.
>
>
>
> >  Q2: For other environments (MPE, GRF, MAMuJoCo), why the performances of MARLlib are only included?
>
> A2: To the best of our knowledge, EPyMARL is the only existing benchmark that unifies diverse MARL algorithms and tasks in one framework, thus is close to MARLlib and suitable to compare with. We cannot provide the result of MPE, GRF, MAMuJoCo from EPyMARL because:
> - we use MPE maintained by PettingZoo. In contrast, EPyMARL uses its own version, which leads to an unfair comparison.
> - GRF & MAMuJoCo are not supported by EPyMARL technically because of their task settings, e.g., continuous action space and 3D observation, while EPyMARL only supports discrete action and 1D input.
>
> Besides, we conduct performance comparisons between different implementations to ensure the correctness of MARLlib. As environments and algorithms are fully independent,  it is reasonable to believe that the implementation is trustworthy if model performance is similar on several famous or most used task sets in MARL, such as SMAC.
>
> However, to further ensure the MARLlib algorithm implementation's trustworthiness and correctness, we conduct extra experiments that compare the performance (reward gain) of MARLlib and the official MAPPO benchmark.  We have incorporated this part of the evaluation and discussion in the paper's Appendix F.
>
>
> >  Q3: Author may clearly illustrate the benefits obtained from agent-level distributed dataflow, a unified agent-environment interface, and effective policy mapping in terms of implementations or experiments.
>
> A3: Thank you very much for the suggestion. We illustrate the benefits of our design both in general (Sec 1) and in detail (Sec 2,3,4).
>
> - Agent-level distributed dataflow helps unify diverse MARL algorithms into one framework.
> - A unified agent-environment interface helps algorithms develop once and run everywhere without environment-wise modification.
> - Effective policy mapping enables one algorithm to tackle different task modes like cooperative, collaborative, and mixed.
>
> All these designs focus on the general unification of MARL. No learning module or function is designed and designated to one typical algorithm, as all algorithms in MARLlib need a fair comparison. Therefore, there is no performance-wise benefit gained. At the same time, efficiency is guaranteed in MARL training as MARLlib is based on the widely used fundamental job scheduling framework Ray. We provide a table of results for your reference. We have incorporated this part of the evaluation and discussion in the paper's Appendix A.
>
>
>
> > Q4: Some statements, like "value iteration used by VDN and QMIX prefers a dense reward function", lack solid explanations.
>
> A4: Thank you for pointing it out. These statements are based on the experiment result from Table 2. VDN and QMIX are significantly worse in sparse reward tasks like GRF. We also conduct experiments on other sparse reward tasks and get similar results. These states reported in EPyMARL: VDN and QMIX perform significantly worse than the policy gradient methods (except COMA). While in tasks with dense reward functions (e.g., SMAC, MPE), VDN and QMIX achieve comparable and robust performance, as shown in Table 2.

---

### Author Response · Authors · 2022-11-16
**Major Revision and Extra Experiments Conducted**

We would like to thank all reviewers for their precious comments, which have helped us enormously in guiding our revision and improvement. In the most respectful manner, we would like to summarize our update as follows.

The revisions we made on MARLlib's codebase and paper:

- fix typos in the instruction files in readme files and documentation
- solution for docker-based running, includes downloadable ready-to-run image and local image build
- tools for website-based convenient training
- paper's revision in the appendix regarding the major concerns and precious suggestions raised by reviewers

The extra experiments we conducted in the rebuttal:

- running efficiency (in clock time) and computation resource usage (GPU memory) comparison between MAPPO, EPyMARL, and MARLlib
- more seeds for data variance analysis
- ablation study on tricks that impact model performance
- performance comparison between MAPPO, EPyMARL, and MARLlib for trustworthy implementation

---

### Decision · Program_Chairs · 2023-01-20

**Decision:**

Reject

**Justification For Why Not Higher Score:**

software framework too early in the development stage

**Justification For Why Not Lower Score:**

N/A

**Metareview: Summary, Strengths And Weaknesses:**

This paper proposes MARLlib, a multi-agent counterpart of the reinforcement learning framework RLlib. The framework supports a wide range of MARL algorithms as well as environments, potentially a very useful platform for replicating baseline method experiments and developing new algorithms.

Reviewers and AC had further discussions on the merit of publication as an ICLR paper at the current stage of development. The challenges with frameworks like this is that they must show some evidence that they will endure the test of time. I think this is the main reason why reviewers were hesitant to give enthusiastic comments. I also think that it’s not too late for the authors to resubmit the paper once MARLlib has gained some degree of popularity among researchers. It’s just too early to have a peer-reviewed archival paper about the software at the current stage.